# Can silicon applied to correct soil acidity in combination with *Azospirillum brasilense* inoculation improve nitrogen use efficiency in maize?

**Fernando Shintate Galindo**[1], **Paulo Humberto Pagliari**[2], **Salatiér Buzetti**[1], **Willian Lima Rodrigues**[1], **José Mateus Kondo Santini**[3], **Eduardo Henrique Marcandalli Boleta**[1], **Poliana Aparecida Leonel Rosa**[1], **Thiago Assis Rodrigues Nogueira**[1], **Edson Lazarini**[1], **Marcelo Carvalho Minhoto Teixeira Filho**[1] *

**1** Department of Plant Health, Rural Engineering, and Soils, São Paulo State University, Ilha Solteira, São Paulo, Brazil, **2** Department of Soil, Water, and Climate, University of Minnesota, Southwest Research and Outreach Center, Lamberton, Minnesota, United States of America, **3** Higher Education Institute of Rio Verde, College Objetivo, Rio Verde, Goiás, Brazil

* mcmtf@yahoo.com.br

**Data Availability Statement:** All relevant data are within the paper and its Supporting Information files.

## Abstract

Alternative management practices are needed to minimize the need for chemical fertilizer use in non-leguminous cropping systems. The use of biological agents that can fix atmospheric N has shown potential to improve nutrient availability in grass crops. This research was developed to investigate if inoculation with *Azospirillum brasilense* in combination with silicon (Si) can enhance N use efficiency (NUE) in maize. The study was set up in a Rhodic Hapludox under a no-till system, in a completely randomized block design with four replicates. Treatments were tested in a full factorial design and included: i) five side dress N rates (0 to 200 kg ha$^{-1}$); ii) two liming sources (Ca and Mg silicate and dolomitic limestone); and iii) with and without seed inoculation with *A. brasilense*. Inoculation with *A. brasilense* was found to increase grain yield by 15% when N was omitted and up to 10% when N was applied. Inoculation also increased N accumulation in plant tissue. Inoculation and limestone application were found to increase leaf chlorophyll index, number of grains per ear, harvest index, and NUE. Inoculation increased harvest index and NUE by 9.5 and 19.3%, respectively, compared with non-inoculated plots. Silicon application increased leaf chlorophyll index and N-leaf concentration. The combination of Si and inoculation provided greater Si-shoot accumulation. This study showed positive improvements in maize growth production parameters as a result of inoculation, but the potential benefits of Si use were less evident. Further research should be conducted under growing conditions that provide some level of biotic or abiotic stress to study the true potential of Si application.

**Funding:** The authors thank FAPESP for the doctoral scholarship (process number: 2017 / 06002-6) by Fernando Shintate Galindo, and CNPq for granting the research productivity scholarship (process number: 312359 / 2017-9) by Prof. Dr. Marcelo Carvalho Minhoto Teixeira Filho.

**Competing interests:** The authors have declared that no competing interests exist.

## Introduction

Maize (*Zea mays* L.) is among of the most relevant and cultivated grain crop used for human and animal consumption [1]. In Brazil, the average yield (e.g., 5,476 kg ha$^{-1}$) is still low compared with high yielding areas of the USA (USDA average 10,840 kg ha$^{-1}$) and Europe (Eurostat average 8,399 kg ha$^{-1}$) [2–5], despite technological improvements in plant nutrition, soil management and field equipment. This low maize yield observed in Brazil is usually attributed to low-fertility soils that require high fertilizer inputs for optimum productivity [6]. Grass crops, such as maize, wheat (*Triticum aestivum*) and rice (*Oryza sativa*), consume approximately 50% of all N-based fertilizer produced [7]. It has been estimated that as much as 15% of the total operating profit of the maize produced in the Brazilian Savannah is due to side-dress N application [8]. It is well known that production and over application of N fertilizer contribute to greenhouse gas emission and water contamination [9–11]. Management practices that improve N use efficiency (NUE) are therefore needed to assure that maize productivity is maximized while reducing the environmental impacts resulting from excessive nutrient application in the Brazilian Savannah [8, 12, 13].

The use of biological agents such as plant growth-promoting bacteria (PGPB) can improve NUE, crop development, and grain yield in cereal crops growing in tropical regions [6, 8, 10]. Several PGPB genera associate with different species of agricultural importance, such as *Azospirillum*, *Arthobacter*, *Azotobacter*, *Bacillus*, *Bradyrhizobium*, *Burkholderia*, *Clostridium*, *Gluconacetobacter*, *Herbaspirillum*, *Pseudomonas*, *Rhizobium* and *Streptomyces* [14]. In Latin America, the use of PGPB in many different crops has increased significantly over the last decade [6, 15]. The *Azospirillum* spp. is one of the most studied plant growth promoting genus [16]. An analysis of field trials conducted worldwide for over 20 years, where various non-legume crops were inoculated with *Azospirillum* spp. under different weather and soil conditions, concluded that crop yield can increase up to 30% with inoculation [17]. Zeffa et al. [14] performed a meta-analysis on the influence of N side dress application associated with *Azospirillum* spp. inoculation in maize and reported an average yield increase of 652 kg ha$^{-1}$.

These bacteria can stimulate plant growth by a series of mechanisms, including the production of phytohormones, such as indole-3-acetic acid (IAA), gibberellins, cytokinins and salicylic acid [18, 19], biological nitrogen fixation (BNF) [20], increase nutrient availability [13] and increase nitric oxide production [21]. In addition, *Azospirillum* spp. has been reported to reduce biotic and abiotic stresses, increase proline content in shoots and roots, improve water potential, increase apoplastic routes for water content, increase cell wall elasticity and chlorophyll content, increase photoprotection pigments and improve stomatal conductance [22]. *Azospirillum* spp. has been reported to increase plant resistance to pathogens, for example promoting disease resistance in rice crops [23] or inhibiting development of bacterial diseases on *Prunus cerasifera* [24]. The level of inoculation response has been described as wide ranging and is determined by plant-inoculant-environment interactions [25].

The use of Si-based fertilizers has been reported to improve N uptake and NUE in different crops [26, 27]. It has been reported that some crops (e.g., sugarcane—*Saccharum officinarum*, rice, wheat, and maize) can absorb Si quickly and in great quantity [28]. In recent years, the number of studies reporting the effects of Si application to crops has increased substantially, especially in grain crops. This increased interest in Si is likely due to the beneficial effects of Si application on plant resistance to abiotic and biotic stresses such as insects and pathogens [29], salt stress and drought [30], and heavy rain and winds [31]. Silicon has also been reported to improve crop yield [32], plant growth, plant architecture, erectness, and photosynthesis rate [33], to decrease transpiration rate [34, 35] and damage caused by pests and diseases [36], and to reduce water use [32]. Recently, Jang et al. [37] concluded that Si application could induce

increased levels of bioactive hormones that participate in various physiological responses. For example, the authors reported an increase in the levels of hormones that are involved in the mitigation of abiotic stress (active endogenous gibberellin GA1, jasmonic acid JA, and salicylic acid SA). Calcium and magnesium silicate also can increase base saturation and extractable levels of P, Ca, Mg, and Si, as well as decrease the phytotoxic effect of Fe, Mn, Zn, Al, and Cd [38, 39] and correct soil acidity [40]. Galindo et al. [41] studying inoculation methods associated with Si application reported an average wheat grain yield increase of 6.7% when seed inoculation and Si application were performed.

The application of Ca and Mg silicate to supply Si in addition to seed inoculation with *A. brasilense* may increase maize grain yield; however, increases in yield are not always observed. Further research with inoculation associated with Si use is needed to determine how to maximize their benefits on NUE, plant development, and productivity. In addition, studies are needed to define how much N fertilizer needs to be applied when Si is applied in combination with *A. brasilense* for optimum grain yield. It would also be valuable to determine if Si utilization has any negative effect when maize is inoculated with *A. brasilense*. The hypothesis of this study was that the application of Si in combination with *A. brasilense* inoculation would improve NUE and reduce the amount of N required for maximum maize production in the Brazilian Savannah.

## Materials and methods

### Field site description

The study was conducted under field conditions in Selvíria (Brazilian Cerrado–Savannah region), state of Mato Grosso do Sul, Brazil ($20°22'S$ and $51°22'W$, 335 m above sea level (a.s. l.)) (S1 Fig), during the crop years of 2015/16 and 2016/17. The soil was classified as Clayey Oxisol (Rhodic Hapludox) according to the Soil Survey Staff [42]. Soil chemical and physical properties were determined from soil samples collected prior to lime application and analysed according to Raij et al. [43]. Total N was determined by the Kjeldahl method [44]. Si was determined after extraction in Ca chloride ($0.01 \text{ mol L}^{-1}$) according to the methodology of Korndörfer et al. [45] (Table 1). Particle size analysis was performed according to Embrapa, [46] and showed the soil at the site used had $471 \text{ g kg}^{-1}$ of sand, $90 \text{ g kg}^{-1}$ of silt and $439 \text{ g kg}^{-1}$ of clay at the 0–0.20 m depth; and $471 \text{ g kg}^{-1}$ of sand, $82 \text{ g kg}^{-1}$ of silt and $447 \text{ g kg}^{-1}$ of clay at the 0.20–0.40 m depth. The experimental area had been cultivated with annual cereal and legume crops for over 30 years. In addition, the area has been under no-tillage for the last 13 years. The crop sequence prior to maize in 2015/16 were maize (2014), soybean (2014/15) and maize (2015) and prior to maize in 2016/17 was wheat (2016). The 2015 maize crop received the application of $180 \text{ kg N ha}^{-1}$. The maximum and minimum temperatures, rainfall, and air relative humidity observed during the study are presented in Fig 1.

### Experimental design, treatment, and field management

The experimental design was a randomized complete block design with four replicates arranged in a $5 \times 2 \times 2$ factorial scheme. There were five N rates applied as side dress (0, 50, 100, 150, and $200 \text{ kg ha}^{-1}$); two liming materials Ca and Mg silicate, which was also the Si source (the composition was 10% of Si, 25% CaO, and 6% MgO) with effective neutralizing power (ENP) of 88; and dolomitic limestone (the composition was 28% CaO, 20% MgO) with an ENP of 80; and two inoculations: with or without seed inoculation with *A. brasilense*. The experimental plots were six 5-m maize rows spaced at a distance of 0.45 m, and the useful area of the plot considered was the central four rows, excluding 0.5 m from each end. Prior to the start of the study a blanket fertilizer application of $375 \text{ kg ha}^{-1}$ of the granular fertilizer 08-

**Table 1. Soil chemical attributes in 0–0.20m and 0.20–0.40m layers before the application of liming sources.**

| Soil chemical attributes | 0–0.20m layer | 0.20m-0.40m layer |
| --- | --- | --- |
| Total N | 1.04 g kg$^{-1}$ | 0.81 g kg$^{-1}$ |
| Si (CaCl$_2$) | 9.4 mg dm$^{-3}$ | 10.2 mg dm$^{-3}$ |
| P (resin) | 19 mg dm$^{-3}$ | 17 mg dm$^{-3}$ |
| S (SO$_4$) | 10 mg dm$^{-3}$ | 30 mg dm$^{-3}$ |
| Organic matter | 21 g dm$^{-3}$ | 16 g dm$^{-3}$ |
| pH (CaCl$_2$) | 5.0 | 4.8 |
| K | 2.1 mmol$_c$ dm$^{-3}$ | 1.2 mmol$_c$ dm$^{-3}$ |
| Ca | 19.0 mmol$_c$ dm$^{-3}$ | 11.0 mmol$_c$ dm$^{-3}$ |
| Mg | 13.0 mmol$_c$ dm$^{-3}$ | 8.0 mmol$_c$ dm$^{-3}$ |
| H+Al | 28.0 mmol$_c$ dm$^{-3}$ | 28.0 mmol$_c$ dm$^{-3}$ |
| Al | 1.0 mmol$_c$ dm$^{-3}$ | 2.0 mmol$_c$ dm$^{-3}$ |
| B (hot water) | 0.17 mg dm$^{-3}$ | 0.11 mg dm$^{-3}$ |
| Cu (DTPA) | 3.1 mg dm$^{-3}$ | 2.1 mg dm$^{-3}$ |
| Fe (DTPA) | 20.0 mg dm$^{-3}$ | 10.0 mg dm$^{-3}$ |
| Mn (DTPA) | 27.2 mg dm$^{-3}$ | 10.7 mg dm$^{-3}$ |
| Zn (DTPA) | 0.8 mg dm$^{-3}$ | 0.2 mg dm$^{-3}$ |
| Cation exchange capacity (pH 7.0) | 62.1 mmol$_c$ dm$^{-3}$ | 48.2 mmol$_c$ dm$^{-3}$ |
| Base saturation (%) | 55 | 42 |

28-16 (N-P$_2$O$_5$-K$_2$O) had to be performed on the entire experimental site to supply phosphorus and potassium. During this blanket nutrient application 30 kg N ha$^{-1}$ was applied to the entire experimental area. Therefore, the total amount of N applied in each treatment is the amount of N applied at the side dress (0 to 200 kg N ha$^{-1}$ as indicated above) in addition to the blanket application of 30kg N ha$^{-1}$. Split application of fertilizers is a general practice used by farmers growing maize in Brazil. Nitrogen treatments were applied manually to evenly distribute the fertilizer on the soil surface without incorporation. The amount of fertilizer needed per plot was applied between the maize rows on December 13, 2015, and December 10, 2016, when the plants were in the V6 stage (with six leaves completely unfolded). After side dress application the experimental area was irrigated with 14 mm of water to minimize ammonia volatilization. Harvest took place on March 15, 2016, and March 21, 2017.

Lime was broadcast applied in one single application at the rate of 1.76 t ha$^{-1}$ for the silicate and 1.94 t ha$^{-1}$ for the limestone 30 days before planting. No incorporation was done as the area was under no-tillage. This is also a common practice used by farmers growing maize in this region of Brazil. The amount of lime applied was based on the amount needed to increase the base saturation to 80% based on the initial soil analysis.

*Azospirillum brasilense* strains Ab-V5 Ab-V6 was inoculated at a rate of 300 mL of liquid inoculant per hectare (guarantee of 2×10$^8$ CFU [colony forming unity] mL$^{-1}$). These are commercial strains used in Brazil with brand name AzoTotal$^{®}$. These strains when used under similar conditions (Brazilian Savannah) have shown positive results in maize development [2, 8, 13, 16]. Seeds were inoculated one hour before planting and after seed treatment with fungicide and insecticide when the seeds were completely dry. The fungicides used were thiophanate-methyl + pyraclostrobin (56 g + 6 g of a.i. per 100 kg of seed) and the insecticide used was fipronil (62 g of a.i. per 100 kg of seed).

Maize DOW 2B710 PW simple hybrid was mechanically sown on November 13 and November 11 for the 15/16 and 16/17 crops, respectively, at the planting density of 3.3 seeds per metre (7.3 plants m$^{-2}$). Seedling emergence occurred five days after sowing, on November

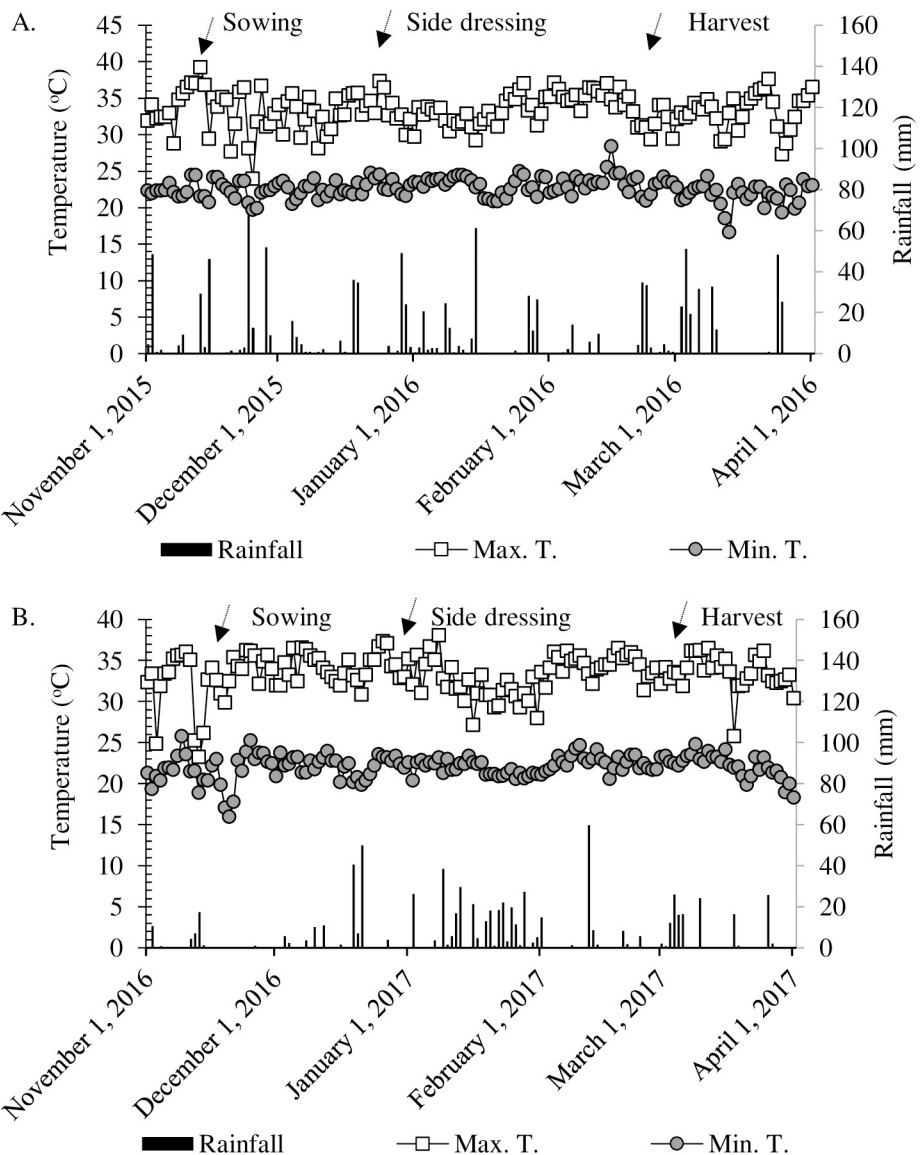

**Fig 1.** Rainfall, air relative humidity, maximum and minimum temperatures obtained from the weather station located in the Education and Research Farm of FE / UNESP during the maize cultivation in the period November 2015 to April 2016 (A) November 2016 to April 2017 (B).

18, 2015, and November 16, 2016, respectively. Supplemental irrigation using a centre pivot sprinkling system was done when need at a water depth of 14 mm. The herbicides atrazine (1000 g ha$^{-1}$ of a.i.) and tembotrione (84 g ha$^{-1}$ of a.i.) in combination with vegetable oil adjuvant (720 g ha$^{-1}$ of a.i.) were used for post-emergence weed control on December 4, 2015 and December 2, 2016, respectively. During the growing season, weeds were controlled by applying 2,4-D (670 g ha$^{-1}$ of the active ingredient [a.i.]) and glyphosate (1800 g ha$^{-1}$ of the a.i.). Insect control was made with triflumuron (24 g ha$^{-1}$ a.i.) and methomyl (215 g ha$^{-1}$ a.i.) on December 20, 2015, and December 17, 2016, respectively. The straw remaining from the previous crop was collected by removing the residue from 10 random points in the experimental area measuring 0.5 m$^2$. The residue sampled was used for chemical tests to determine nutrient accumulation (Table 2).

**Table 2. Nutrient accumulation in maize straw (2015/16 predecessor crop).**

| N | P | K | Ca | Mg | S | Si | B | Cu | Fe | Mn | Zn | C/N ratio |
|---|---|---|---|----|---|----|---|----|----|----|----|-----------|
| ------------ (kg ha$^{-1}$)------------ | | | | | | | ----------- (g ha$^{-1}$) ----------- | | | | | |
| 78.2 | 7.2 | 68.8 | 23.3 | 21.1 | 16.3 | 13.1 | 260.4 | 74.0 | 1018.1 | 709.6 | 185.1 | 38.3 |

## Measurements collected

**In season evaluations.** During the growing season, leaf chlorophyll index (LCI) was indirectly measured in 10 plants using a portable non-destructive chlorophyll meter ClorofiLOG® model CFL-1030 [47]. Tissue N and Si concentration was determined by collecting 20 leaves from the main ear insertion during the female flowering stage from each plot. The leaves were then used to determine N and Si uptake by tissue. Nitrogen and Si concentration in tissue was determined according to Cantarella et al. [48]. In addition, N and Si concentration in shoot and roots were also quantified, and N analysis followed the methodology proposed by Malavolta et al. [49], while Si analysis followed the methodology proposed by Silva [50].

**Productive components, NUE, and grain yield.** Root and shoot dry matter were measured during the female flowering stage in each experimental plot by collecting 5 plants per plot. Plant height and diameter were measured at maturity in 10 plants per plot. Height was measured from the ground surface to the apex of the tassel, and stem diameter was measured in the second internode of the plant using a caliper. Ten ears were collected at harvest to determine ear diameter, ear length (determined from the base of the ear to the apex), number of rows per ear, number of grains per row, number of grains per ear, and mass of 100 grains adjusted to 13% moisture. Harvest index (HI) and nitrogen use efficiency (NUE) wer then calculated with the Eqs 1 and 2 (below). Grain yield was determined by harvesting the useful experimental area and adjusted to 13%.

$$\text{HI, expressed as \% : } [\text{Grain Yield} \div (\text{Grain Yield} + \text{Shoot Yield})] \times 100 \qquad \text{Eq1}$$

$$\text{NUE, according to Moll et al.}[51] : [(\text{GYF} - \text{GYW}) \div (\text{amount of N applied})] \qquad \text{Eq2}$$

Where, GYF = Grain Yield with fertilizer and GYW = Grain Yield without fertilizer

## Statistical analysis

All data were initially tested for normality using the Shapiro and Wilk [52] test which showed the data to be normally distributed ($W \geq 0.90$). The data was analysed by ANOVA in a 3-way factorial design with N application rates, liming material source and inoculation and their interactions considered fixed effects in the model using ExpDes package. Mean separation was done when significant factors or interactions were observed using the test Tukey. Regression analysis was used to discern whether there was a linear or non-linear response to N rates using R [53].

## Results

### Leaf chlorophyll index and plant nutrient uptake

Nitrogen leaf concentration responded linearly to N application rates in 2015/16 (Tables 3 and 4). In 2016/17, in the absence of inoculation, the application of limestone resulted in greater N-leaf concentration compared to silicate application; however, with inoculation, silicate provided greater concentration compared to limestone use (Tables 3 and 5). In 2016/17, the control treatment (0 N) showed greater N concentration when inoculated compared with non-

**Table 3. *P*-values for N and Si-leaf concentration, leaf chlorophyll index, N and Si-shoot and root accumulation, shoot and root dry matter, plant height, stem diameter, ear length and diameter, number of rows per ear, grains per row and grains per ear, mass of 100 grains, harvest index, nitrogen use efficiency, and corn grain yield affected by rates of nitrogen, liming sources, with or without inoculation with *Azospirillum brasilense*.** 2015/16 and 2016/17.

| | N-leaf concentration | | Si-leaf concentration | | Leaf chlorophyll index | | N-shoot accumulation | |
| --- | --- | --- | --- | --- | --- | --- | --- | --- |
| | ---------- (g kg$^{-1}$ of D. M.) --------- | | | | | | --------- (kg ha$^{-1}$) --------- | |
| *P*-values | 2015/2016 | 2016/2017 | 2015/2016 | 2016/2017 | 2015/2016 | 2016/2017 | 2015/2016 | 2016/2017 |
| Rates (R) | 0.009** | 0.001** | 0.949$^{ns}$ | 0.026* | 0.643$^{ns}$ | 0.001** | 0.026* | 0.009** |
| Sources (S) | 0.596$^{ns}$ | 0.594$^{ns}$ | 0.001** | 0.007** | 0.002** | 0.032* | 0.300$^{ns}$ | 0.209$^{ns}$ |
| Inoculation (I) | 0.915$^{ns}$ | 0.056$^{ns}$ | 0.062$^{ns}$ | 0.648$^{ns}$ | 0.009** | 0.041* | 0.465$^{ns}$ | 0.606$^{ns}$ |
| R × S | 0.231$^{ns}$ | 0.942$^{ns}$ | 0.135$^{ns}$ | 0.268$^{ns}$ | 0.002** | 0.001** | 0.765$^{ns}$ | 0.980$^{ns}$ |
| R × I | 0.364$^{ns}$ | 0.001** | 0.942$^{ns}$ | 0.946$^{ns}$ | 0.955$^{ns}$ | 0.631$^{ns}$ | 0.043* | 0.794$^{ns}$ |
| R × I | 0.914$^{ns}$ | 0.001** | 0.172$^{ns}$ | 0.197$^{ns}$ | 0.008** | 0.875$^{ns}$ | 0.855$^{ns}$ | 0.183$^{ns}$ |
| R × S × I | 0.342$^{ns}$ | 0.135$^{ns}$ | 0.604$^{ns}$ | 0.776$^{ns}$ | 0.311$^{ns}$ | 0.739$^{ns}$ | 0.431$^{ns}$ | 0.870$^{ns}$ |

| | Si-shoot accumulation | | N-root accumulation | | Si-root accumulation | | Shoot dry matter | |
| --- | --- | --- | --- | --- | --- | --- | --- | --- |
| | ------------——— (kg ha$^{-1}$) ------------—— | | | | | | | |
| *P*-values | 2015/2016 | 2016/2017 | 2015/2016 | 2016/2017 | 2015/2016 | 2016/2017 | 2015/2016 | 2016/2017 |
| Rates (R) | 0.10$^{ns}$ | 0.008** | 0.001** | 0.001** | 0.001** | 0.015* | 0.116$^{ns}$ | 0.044* |
| Sources (S) | 0.935$^{ns}$ | 0.311$^{ns}$ | 0.351$^{ns}$ | 0.513$^{ns}$ | 0.001** | 0.001** | 0.763$^{ns}$ | 0.299$^{ns}$ |
| Inoculation (I) | 0.577$^{ns}$ | 0.553$^{ns}$ | 0.356$^{ns}$ | 0.575$^{ns}$ | 0.949$^{ns}$ | 0.538$^{ns}$ | 0.506$^{ns}$ | 0.801$^{ns}$ |
| R × S | 0.997$^{ns}$ | 0.749$^{ns}$ | 0.012* | 0.014* | 0.139$^{ns}$ | 0.242$^{ns}$ | 0.986$^{ns}$ | 0.868$^{ns}$ |
| R × I | 0.559$^{ns}$ | 0.378$^{ns}$ | 0.015* | 0.004** | 0.707$^{ns}$ | 0.622$^{ns}$ | 0.666$^{ns}$ | 0.535$^{ns}$ |
| R × I | 0.384$^{ns}$ | 0.004** | 0.692$^{ns}$ | 0.508$^{ns}$ | 0.052$^{ns}$ | 0.695$^{ns}$ | 0.242$^{ns}$ | 0.085$^{ns}$ |
| R × S × I | 0.472$^{ns}$ | 0.810$^{ns}$ | 0.648$^{ns}$ | 0.804$^{ns}$ | 0.309$^{ns}$ | 0.221$^{ns}$ | 0.391$^{ns}$ | 0.903$^{ns}$ |

| | Root dry matter | | Plant height | | Stem diameter | | Ear lenght | |
| --- | --- | --- | --- | --- | --- | --- | --- | --- |
| | ———— (kg ha$^{-1}$) ——— | | ——— (m) ——— | | ———— (cm)) ——— | | | |
| *P*-values | 2015/2016 | 2016/2017 | 2015/2016 | 2016/2017 | 2015/2016 | 2016/2017 | 2015/2016 | 2016/2017 |
| Rates (R) | 0.001** | 0.044* | 0.444$^{ns}$ | 0.001** | 0.006** | 0.001** | 0.001** | 0.001** |
| Sources (S) | 0.041* | 0.773$^{ns}$ | 0.240$^{ns}$ | 0.523$^{ns}$ | 0.016* | 0.530$^{ns}$ | 0.107$^{ns}$ | 0.332$^{ns}$ |
| Inoculation (I) | 0.299$^{ns}$ | 0.592$^{ns}$ | 0.084$^{ns}$ | 0.711$^{ns}$ | 0.267$^{ns}$ | 0.737$^{ns}$ | 0.142$^{ns}$ | 0.234$^{ns}$ |
| R × S | 0.002** | 0.521$^{ns}$ | 0.869$^{ns}$ | 0.614$^{ns}$ | 0.113$^{ns}$ | 0.139$^{ns}$ | 0.567$^{ns}$ | 0.280$^{ns}$ |
| R × I | 0.096$^{ns}$ | 0.785$^{ns}$ | 0.864$^{ns}$ | 0.960$^{ns}$ | 0.064$^{ns}$ | 0.475$^{ns}$ | 0.177$^{ns}$ | 0.573$^{ns}$ |
| R × I | 0.324$^{ns}$ | 0.537$^{ns}$ | 0.586$^{ns}$ | 0.088$^{ns}$ | 0.377$^{ns}$ | 0.001** | 0.385$^{ns}$ | 0.270$^{ns}$ |
| R × S × I | 0.358$^{ns}$ | 0.598$^{ns}$ | 0.886$^{ns}$ | 0.573$^{ns}$ | 0.236$^{ns}$ | 0.267$^{ns}$ | 0.228$^{ns}$ | 0.678$^{ns}$ |

| | Ear diameter | | Number of rows per ear | | Number of grains per row | | Number of grains per ear | |
| --- | --- | --- | --- | --- | --- | --- | --- | --- |
| | --------- (cm) --------- | | | | | | | |
| *P*-values | 2015/2016 | 2016/2017 | 2015/2016 | 2016/2017 | 2015/2016 | 2016/2017 | 2015/2016 | 2016/2017 |
| Rates (R) | 0.386$^{ns}$ | 0.003** | 0.733$^{ns}$ | 0.919$^{ns}$ | 0.225$^{ns}$ | 0.001** | 0.001** | 0.005** |
| Sources (S) | 0.640$^{ns}$ | 0.013* | 0.222$^{ns}$ | 0.306$^{ns}$ | 0.339$^{ns}$ | 0.668$^{ns}$ | 0.746$^{ns}$ | 0.438$^{ns}$ |
| Inoculation (I) | 0.932$^{ns}$ | 0.233$^{ns}$ | 0.812$^{ns}$ | 0.011* | 0.794$^{ns}$ | 0.190$^{ns}$ | 0.023* | 0.014* |
| R × S | 0.392$^{ns}$ | 0.515$^{ns}$ | 0.086$^{ns}$ | 0.734$^{ns}$ | 0.152$^{ns}$ | 0.571$^{ns}$ | 0.121$^{ns}$ | 0.930$^{ns}$ |
| R × I | 0.124$^{ns}$ | 0.201$^{ns}$ | 0.898$^{ns}$ | 0.200$^{ns}$ | 0.344$^{ns}$ | 0.110$^{ns}$ | 0.015* | 0.116$^{ns}$ |
| R × I | 0.545$^{ns}$ | 0.894$^{ns}$ | 0.504$^{ns}$ | 0.309$^{ns}$ | 0.606$^{ns}$ | 0.054$^{ns}$ | 0.299$^{ns}$ | 0.340$^{ns}$ |
| R × S × I | 0.632$^{ns}$ | 0.572$^{ns}$ | 0.901$^{ns}$ | 0.329$^{ns}$ | 0.171$^{ns}$ | 0.441$^{nsns}$ | 0.339$^{ns}$ | 0.198$^{ns}$ |

| | Mass of 100 grains | | Harvest index | | Nitrogen use efficiency | | Grain Yield | |
| --- | --- | --- | --- | --- | --- | --- | --- | --- |
| | ---------- (g) ---------- | | ------- (%) ------- | | -- (kg grains kg N applied$^{-1}$) -- | | ----- (kg ha$^{-1}$) ----- | |
| *P*-values | 2015/2016 | 2016/2017 | 2015/2016 | 2016/2017 | 2015/2016 | 2016/2017 | 2015/2016 | 2016/2017 |
| Rates (R) | 0.002** | 0.001** | 0.045* | 0.009** | 0.001** | 0.001** | 0.001** | 0.001** |
| Sources (S) | 0.010* | 0.260$^{ns}$ | 0.031* | 0.258$^{ns}$ | 0.918$^{ns}$ | 0.467$^{ns}$ | 0.840$^{ns}$ | 0.995$^{ns}$ |
| Inoculation (I) | 0.081$^{ns}$ | 0.234$^{ns}$ | 0.027* | 0.004** | 0.007** | 0.193$^{ns}$ | 0.006** | 0.033* |
| R × S | 0.941$^{ns}$ | 0.211$^{ns}$ | 0.920$^{ns}$ | 0.667$^{ns}$ | 0.249$^{ns}$ | 0.001$^{ns}$ | 0.008** | 0.001** |

*(Continued)*

**Table 3.** (Continued)

| | | | | | | | | |
|---|---|---|---|---|---|---|---|---|
| R × I | $0.608^{ns}$ | $0.509^{ns}$ | $0.708^{ns}$ | $0.562^{ns}$ | $0.422^{ns}$ | $0.489^{ns}$ | $0.003^{**}$ | $0.001^{**}$ |
| R × I | $0.445^{ns}$ | $0.220^{ns}$ | $0.748^{ns}$ | $0.482^{ns}$ | $0.380^{ns}$ | $0.752^{ns}$ | $0.118^{ns}$ | $0.324^{ns}$ |
| R × S × I | $0.588^{ns}$ | $0.528^{ns}$ | $0.616^{ns}$ | $0.589^{ns}$ | $0.063^{ns}$ | $0.122^{ns}$ | $0.115^{ns}$ | $0.781^{ns}$ |

$^{**}$, $^{*}$ and ns: significant at $p<0.01$, $p<0.05$, and not significant, respectively.

inoculated treatments (Fig 2A). Furthermore, inoculation changed the behaviour of N concentration in tissue (Table 3). Nitrogen-leaf concentration in tissue responded linearly to N application rates in the presence of inoculation and non-linearly in the absence of inoculation (Fig 2A). Si-leaf concentration responded linearly to N application rates (Tables 3 and 4) and was greater in the plots with Ca and Mg silicate application in both years than in limestone amended plots (Tables 3 and 6).

LCI response to N application changed based on the liming materials used and also in the cropping season (Table 3, Fig 2B and 2C). In 2015/16, LCI did not respond to N application when limestone was used, while it showed a non-linear response to increasing N rates when Ca Mg silicate was used. The highest LCI measured was observed when 93 kg N ha$^{-1}$ was applied (Fig 2B). In 2016/17, LCI response to N application was found to be non-linear for both liming sources with Ca and Mg silicate tending to provide higher LCI at lower N

**Table 4. 2015/2016 and 2016/2017 crop season significant regression equations as a function of N rates.**

| Independent variable | 2015/16 crop season |
|---|---|
| N-leaf concentration | $\hat{Y} = 28.1655 + 0.0077x \ (R^2 = 0.84^{**})$ |
| Si-root accumulation | $\hat{Y} = 5.1999 + 0.0683x - 0.0002x^2 \ (R^2 \ 0.99^{**})$ |
| Stem diameter | $\hat{Y} = 2.0797 + 0.0004x \ (R^2 = 0.71^{**})$ |
| Ear length | $\hat{Y} = 16.8721 + 0.0043x \ (R^2 = 0.71^{**})$ |
| Mass of 100 grains | $\hat{Y} = 30.3150 + 0.0079x \ (R^2 = 0.98^{**})$ |
| Harvest index | $\hat{Y} = 41.0715 + 0.0236x \ (R^2 = 0.55^{*})$ |
| Nitrogen use efficiency | $\hat{Y} = 20.7759 - 0.0592x \ (R^2 = 0.87^{**})$ |
| Independent variable | 2016/2017 crop season |
| Si-leaf concentration | $\hat{Y} = 16.8082 + 0.0070x \ (R^2 = 0.65^{*})$ |
| N-shoot accumulation | $\hat{Y} = 152.5325 + 0.3496x \ (R^2 = 0.99^{**})$ |
| Si-shoot accumulation | $\hat{Y} = 24.5345 + 0.0707x \ (R^2 = 0.96^{**})$ |
| Si-root accumulation | $\hat{Y} = 6.9308 + 0.0146x \ (R^2 = 0.54^{*})$ |
| Shoot dry matter | $\hat{Y} = 10123.4143 + 15.0346x \ (R^2 = 0.96^{*})$ |
| Root dry matter | $\hat{Y} = 768.6313 + 1.2807x \ (R^2 = 0.73^{*})$ |
| Plant height | $\hat{Y} = 2.4589 + 0.0019x - 0.000005x^2 \ (R^2 = 0.95^{**})$ |
| Stem diameter | $\hat{Y} = 2.0323 + 0.0007x \ (R^2 = 0.90^{**})$ |
| Ear length | $\hat{Y} = 15.3472 + 0.0126x \ (R^2 = 0.92^{**})$ |
| Ear diameter | $\hat{Y} = 5.4497 + 0.0014x \ (R^2 = 0.78^{**})$ |
| Number of grains per row | $\hat{Y} = 32.9052 + 0.0243x \ (R^2 = 0.92^{**})$ |
| Number of grains per ear | $\hat{Y} = 605.9887 + 0.3856x \ (R^2 = 0.89^{**})$ |
| Mass of 100 grains | $\hat{Y} = 30.0132 + 0.0141x \ (R^2 = 0.97^{**})$ |
| Harvest index | $\hat{Y} = 42.2123 + 0.0269x \ (R^2 = 0.79^{**})$ |
| Nitrogen use efficiency | $\hat{Y} = 26.2053 - 0.0787x \ (R^2 = 0.94^{**})$ |

$^{**}$ and $^{*}$: significant at $p<0.01$ and $p<0.05$, respectively.

**Table 5. Interaction between inoculation and liming sources in leaf chlorophyll index, N-leaf concentration, Si-shoot accumulation and stem diameter.** 2015/16 and 2016/17.

| Source | 2015/16 crop season | |
|---|---|---|
| | Limestone | Silicate |
| | **Leaf chlorophyll index** | |
| With *A. brasilense* | 69 aA ± 2.9† | 71 aA ± 2.1 |
| Without *A. brasilense* | 66 bB ± 3.0 | 70 aA ± 3.6 |
| | 2016/17 crop season | |
| | N-leaf concentration (g kg$^{-1}$) | |
| With *A. brasilense* | 27 bB ± 2.9 | 29 aA ± 2.8 |
| Without *A. brasilense* | 28 aA ± 2.4 | 26 bB ± 3.1 |
| | Si-shoot accumulation (kg ha$^{-1}$) | |
| With *A. brasilense* | 23 bB ± 0.7 | 38 aA ± 1.1 |
| Without *A. brasilense* | 36 aA ± 1.0 | 28 aA ± 0.5 |
| | Stem diameter (cm) | |
| With *A. brasilense* | 2.0 bB ± 0.07 | 2.1 aA ± 0.09 |
| Without *A. brasilense* | 2.1 aA ± 0.08 | 2.0 aB ± 0.09 |

†The letters correspond to a significant difference at 5% probability level ($p \leq 0.05$). Uppercase letters indicate difference between inoculation or not with *A. brasilense*, and lowercase letters indicate differences between liming sources, respectively. Means are followed by the standard deviation ($n = 4$).

application rates (Fig 2C). In addition, in 2015/16, in the absence of inoculation, the use of silicate led to higher LCI levels compared to the use of limestone (Tables 3 and 5). Also, in 2016/17 inoculation resulted in greater LCI compared to non-inoculated plots (Tables 3 and 7).

In 2015/16, N shoot accumulation tended to respond linearly to N application rates when plots were inoculated with no trends for non-inoculated plots (Table 3, Fig 2D). In 2016/17, N-shoot accumulation respond linearly to N application regardless of inoculation (Tables 3 and 4). It was observed that in 2016/17 the application of limestone increased Si-shoot accumulation in non-inoculated plots (Tables 3 and 5). Silicon accumulation in tissue was found to respond linearly to N application rates in 2016/17 (Tables 3 and 4).

Nutrient accumulation in the root varied based on N application rates, liming materials, inoculation, and cropping seasons (Table 3). In 2015/16, N-root accumulation was found to respond linearly to N application rates when limestone was used and non-linearly with the application of Ca and Mg silicate (Table 3). It was observed that the highest N accumulation was achieved when 143 kg N ha$^{-1}$ was applied in combination with Ca and Mg silicate (Fig 2E). Contrasting results were observed for the 2016/17 season; the N accumulation in root response to N application rates was linear for the Ca and Mg silicate and non-linear for the limestone (Fig 2F). Although, the highest accumulation was observed when Ca and Mg silicate was used, root N accumulation tended to be greater when plots were treated with limestone (Fig 2E and 2F). Similarly, N accumulation in root response to N rates showed contrasting results under the different inoculation treatments (Fig 3A and 3B). Root N accumulation as a function of N application rates in plots receiving inoculation was non-linear in the 2015/16 season and linear in the 2016/17 season (Fig 3A and 3B). In contrast, root N accumulation as a function of N application rates in non-inoculated plots was linear in the 2015/16 season and non-linear in the 2016/17 season (Fig 3A and 3B). Although there were significant differences, the results of the study do not allow for a clear understanding of the effects of liming materials and inoculation on nutrient accumulation in maize roots. Si-root accumulation showed different responses to N application rates and was found to respond non-linearly in 2015/16 and

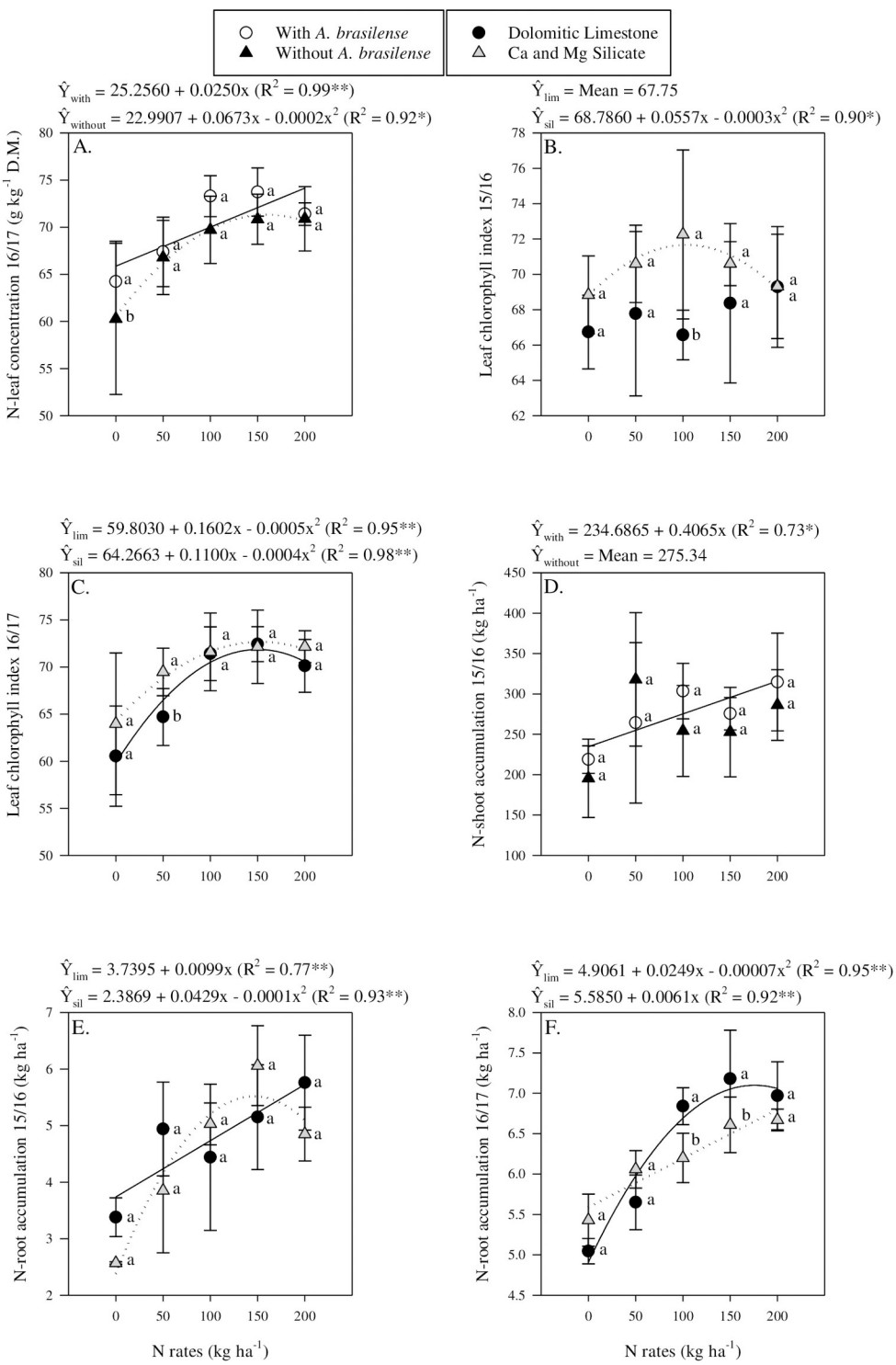

**Fig 2.** Interaction between inoculation and nitrogen rates in N-leaf concentration in 2016/17 (a), interaction between liming sources and nitrogen rates in LCI in 2015/16 (b) and 2016/17 (c), interaction between inoculation and nitrogen rates in N-shoot accumulation in 2015/16 (d), interaction between liming sources and nitrogen rates in N-root accumulation in 2015/16 (e) and 2016/17 (f). The letters correspond to a significant difference at 5% probability level ($p \leq 0.05$). $^{**}$ and $^{*}$: significant at $p<0.01$ and $p<0.05$, respectively. Error bars indicate the standard deviation of the mean ($n = 4$). P.M. = point of maximum response to N rates.

**Table 6. Si-leaf concentration, Si-root accumulation, stem diameter, mass of 100 grains, ear diameter and harvest index as a function of liming sources.** 2015/16 and 2016/17.

| Source | 2015/16 crop season | 2016/17 crop season |
|---|---|---|
| | Si-leaf concentration (g kg$^{-1}$) | |
| Limestone | 11.1 b ± 1.4† | 16.9 b ± 3.4 |
| Silicate | 12.7 a ± 1.5 | 18.1 a ± 4.2 |
| | Si-root accumulation (kg ha$^{-1}$) | |
| Limestone | 7.6 b ± 2.2 | 6.6 b ± 2.5 |
| Silicate | 9.7 a ± 2.9 | 10.1 a ± 2.4 |
| | Stem diameter (cm) | |
| Limestone | 2.1 b ± 0.09 | - |
| Silicate | 2.2 a ± 0.10 | - |
| | Mass of 100 grains (g) | |
| Limestone | 31 b ± 1.3 | - |
| Silicate | 32 a ± 1.1 | - |
| | Ear diameter (cm) | |
| Limestone | - | 5.7 a ± 0.2 |
| Silicate | - | 5.5 b ± 0.2 |
| | Harvest index (%) | |
| Limestone | 45 a ± 4.6 | - |
| Silicate | 41 b ± 5.1 | - |

†The letters correspond to a significant difference at 5% probability level ($p \leq 0.05$). Error bars indicate the standard deviation of the mean ($n = 4$).

linearly in 2016/17 (Tables 3 and 4). In general, Si-root accumulation was greater in the plots with Ca and Mg silicate application in both years (Tables 3 and 6).

**Table 7. Leaf chlorophyll index, number of rows per ear, grains per ear, harvest index, and N use efficiency as a function of inoculation.** 2015/16 and 2016/17.

| Source | 2015/16 crop season | 2016/17 crop season |
|---|---|---|
| | Leaf chlorophyll index | |
| With *A. brasilense* | - | 70 a ±2.9† |
| Without *A. brasilense* | - | 67 b ± 2.7 |
| | Number of rows per ear | |
| With *A. brasilense* | - | 18.5 a ± 0.7 |
| Without *A. brasilense* | - | 17.9 b ± 0.6 |
| | Number of grains per ear | |
| With *A. brasilense* | - | 662 a ± 52 |
| Without *A. brasilense* | - | 627 b ± 70 |
| | Harvest index (%) | |
| With *A. brasilense* | 45 a ± 5.3 | 47 a ± 3.8 |
| Without *A. brasilense* | 42 b ± 4.1 | 43 b ± 4.7 |
| | N use efficiency (kg grains kg N applied$^{-1}$) | |
| With *A. brasilense* | 14.8 a ± 3.6 | - |
| Without *A. brasilense* | 11.9 b ± 3.1 | - |

†The letters correspond to a significant difference at 5% probability level ($p \leq 0.05$). Error bars indicate the standard deviation of the mean ($n = 4$).

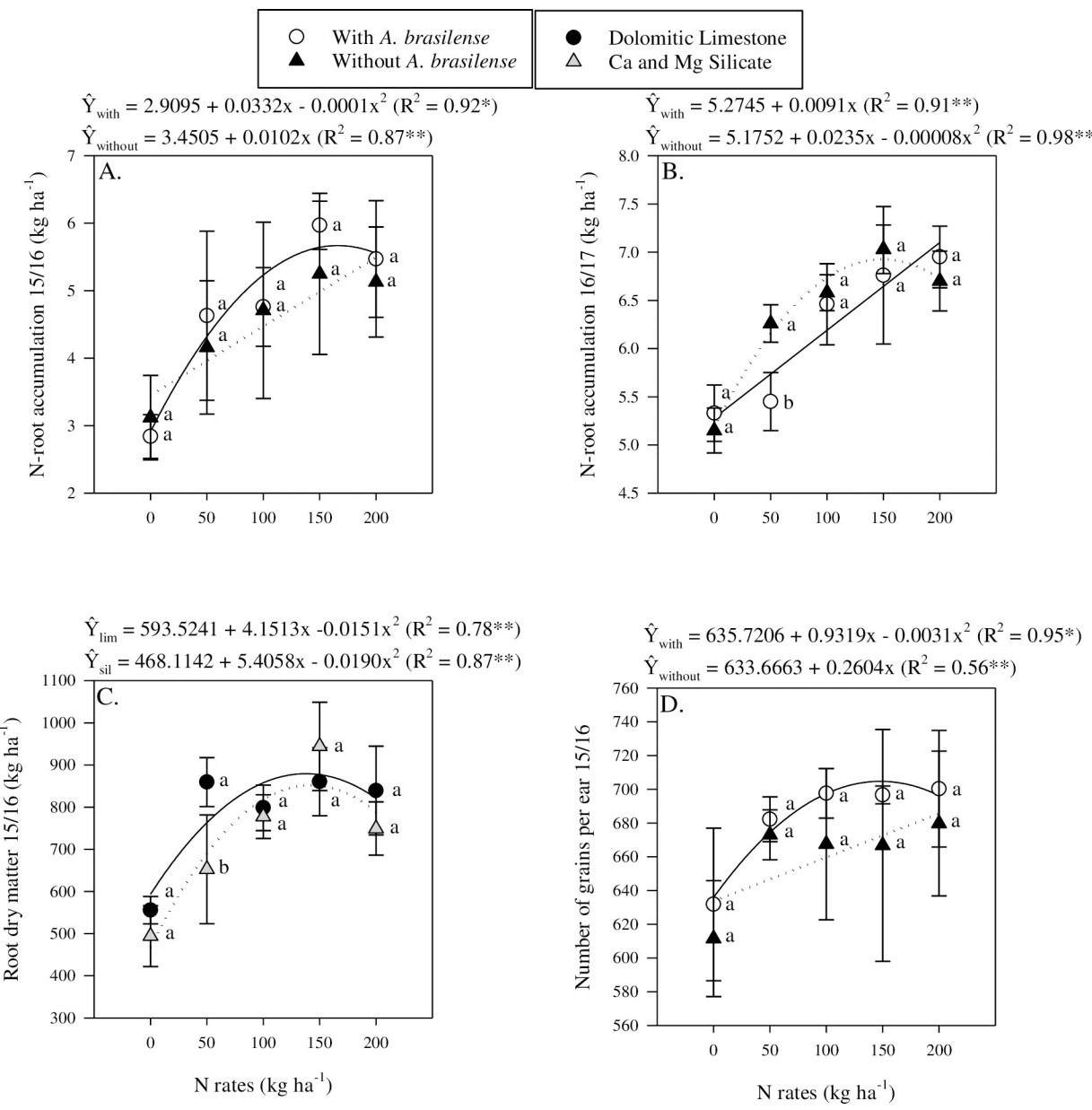

**Fig 3.** Interaction inoculation and nitrogen rates in N-root accumulation in 2015/16 (a) and 2016/17 (b), interaction between liming sources and nitrogen rates in root dry matter in 2015/16 (c) and in number of grains per ear in 2015/16 (d). The letters correspond to a significant difference at 5% probability level ($p \leq 0.05$). ** and *: significant at $p<0.01$ and $p<0.05$, respectively. Error bars indicate the standard deviation of the mean ($n = 4$). P.M. = point of maximum response to N rates.

## Productive components and NUE

In 2015/16, root biomass dry matter showed a non-linear response to N application rates and the application of limestone tended to result in greater root dry matter accumulation than Ca and Mg silicate, but only at low N application rates (Table 3, Fig 3C). In the second season, root biomass dry matter was found to respond linearly to N rates regardless of inoculation of silicate sources (Tables 3 and 4).

In 2016/17, inoculation had opposite effects on stem diameter based on liming materials (Tables 3 and 5). Stem diameter increased under inoculation when Ca and Mg silicate was used compared with non-inoculated; in contrast, stem diameter was greater in non-inoculated plots when limestone was used (Tables 3 and 5). In both years, stem diameter showed a linear response to N rates (Tables 3 and 4).

The number of grains per ear response to N application rates showed different behaviour based on inoculation in the 2015/16 season (Table 3). In 2015/16, the number of grains per ear in plots receiving inoculation showed a non-linear response to N application rates while non-inoculated plots showed a linear response to N rates (Fig 3D). Although there were different responses based on inoculation, there was no clear evidence of the effect of inoculation on the number of grains per ears in the first season. In 2016/17, the number of grains per ear was found to respond linearly to N application rates and also was greater when plots were inoculated compared with non-inoculated plots (Tables 3, 4 and 7).

Ear length, mass of 100 grains and harvest index were found to respond linearly to increasing N rates in both cropping years (Tables 3 and 4). Shoot dry matter, ear diameter and number of grains per row were found to respond linearly to increasing N rates in 2016/17 but showed no response in 2015/16 (Tables 3 and 4). Also, in 2016/17, plant height showed a non-linear response to increasing N application rates (Tables 3 and 4). In contrast, NUE was found to decrease linearly to increasing N application rates in both years (Tables 3 and 4). The use of Ca and Mg silicate lead to greater 100 grain mass in 2015/16; however, this liming source decreased ear diameter in 2016/17 and harvest index in 2015/16 (Tables 3 and 6). Inoculation with *A. brasilense* lead to greater harvest index in both years, NUE in 2015/16 (2.84 kg grains kg N applied$^{-1}$, an increase that is equivalent to 23.7%), and number of rows per ear in 2016/17 compared with non-inoculated plots (Tables 3 and 7).

## Grain yield

Maize grain yield showed different behaviours in the two years of the study based on liming source (Table 3). In the 2015/16 season, application of Ca and Mg silicate caused a non-linear response to N application rates, while the application of limestone led to a linear response to N application rate (Fig 4A). In 2016/17, maize grain yield responded linearly to N rates, with plots treated with limestone showing a faster rate of increase in grain yield per kg of N applied than plots treated with Ca and Mg silicate (Fig 4B).

The effect of inoculation and N application rates on grain yield was similar to what was observed for liming source and N application rates (Table 3). In the first season, plants receiving inoculation showed a linear response to increasing N rate, while non-inoculated plants showed a non-linear response to N rates (Fig 4C). Although different responses to N rates were observed for the two inoculation treatments, the most significant difference was observed at the highest N rate (Fig 4C). In 2016/17, plots that did not receive N yielded 7,955 kg ha$^{-1}$ when inoculated and 6,905 kg ha$^{-1}$ when not inoculated (Fig 4D). In addition, in 2016/17 grain yield was found to respond linearly to N application rates with non-inoculated plots having a greater rate of increase per kg of N applied than inoculated plots (Fig 4D). The results of the study showed that in 2016/17 each kg of N added resulted in an increase of 7.6 kg of grain for inoculated plots and 12.9 kg of grain for non-inoculated plots (Fig 4D). The lower rate of response observed for inoculated plots likely reflects the fact that inoculation provided some N to the maize during the growing season.

## Discussion

N is the nutrient that is most demanded by maize plants and directly affects crop development and yield. The higher N availability as a function of the N application likely favoured the

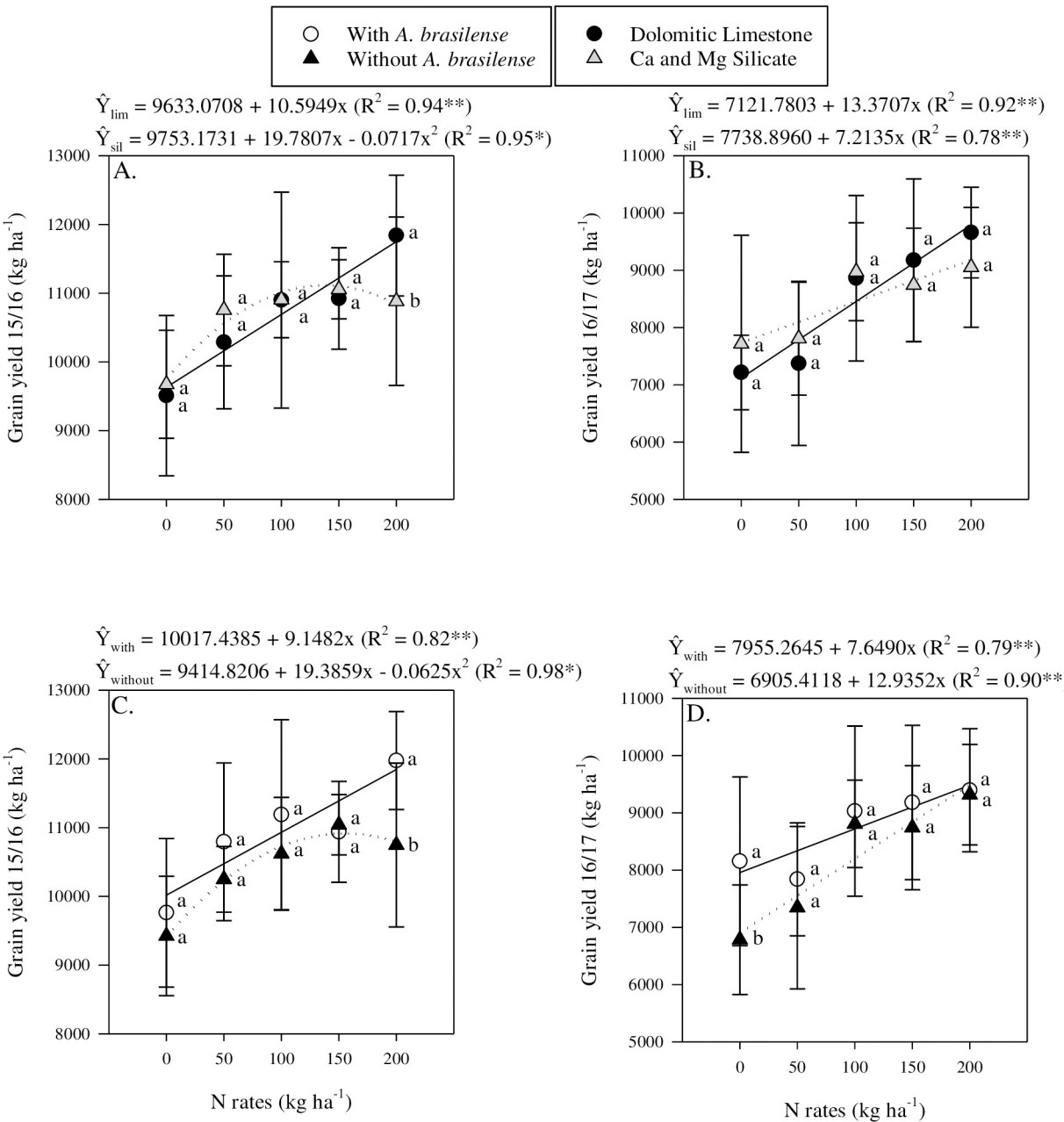

**Fig 4.** Interaction between liming sources and nitrogen rates in grain yield in 2015/16 (a) and 2016/17 (b). Interaction between inoculation and nitrogen rates in grain yield in 2015/16 (c) and 2016/17 (d). The letters correspond to a significant difference at 5% probability level ($p \leq 0.05$). ** and *: significant at $p < 0.01$ and $p < 0.05$, respectively. Error bars indicate the standard error of the mean ($n = 4$). P.M. = point of maximum response to N rates.

development of a well established root system leading to improved shoot development and maize grain yield. Increased root dry matter can positively influence root scavenging, that is important for the interception of N and Si in crop systems. Although the exact mechanisms underlying the PGPB effect on N acquisition by maize was not evaluated in the present study, it is possible that the greater NUE observed in inoculated plants was due to its ability to promote plant growth [6, 16, 19, 25]. Specifically, it has been demonstrated in draft genome

sequences that the strains Ab-V5 and Ab-V6 of *A. brasilense*, carry similar *nif* and *fix* genes that confer their ability to fix atmospheric N [54]. Although the strains differ in their capacity to synthesize phytohormones [19, 55], both share the same genes related to the synthesis of auxins. This growth promotion mechanism might have improved the ability of plants to explore the soil more efficiently, as indicated in previous studies using *A. brasilense* [6, 55–59]. Also, according to Cormier et al. [60], two strategies may be devised for N use efficiency improvement: increasing the yield at a constant N supply and/or maintaining high yield when reducing N supply. In the first season, our data showed that grain yield in inoculated plots at the highest N rate tested yielded 10% more than non-inoculated plots. In the second season, a different behaviour was observed and a residual effect or NBF was likely observed (Fig 4D). Plots that were inoculated showed a slower rate of response to applied N compared with plots that were not inoculated. This was very evident in the control treatment where no N was added and yields were 15% greater in the inoculated plots. The lower rate of response for grain yield to N applied in inoculated plots suggests that N was being supplied from sources other than the fertilizer added. Similar results were reported elsewhere when side dress N application was found to improve maize yield between 3.8 to 27% in inoculated plants compared with non-inoculated plants [16, 61–63]. The results of our research show that inoculation with *A. brasilense* has potential as a strategy to improve NUE.

In this study, most of the Si taken up by the plant accumulated in the leaf tissue because Si deposits are known to occur more frequently in tissues where water is lost in large amounts due to plant transpiration [64, 65]. Silicon translocation rate varies among plant species, but once deposited in the cell wall it becomes immobile [66]. Some grass species, such as maize, can take up and redistribute large amounts of Si in the aboveground tissue due to specific Si transporters [67]. The amount of Si accumulated in the roots was greater than the amount of N accumulated in the roots. The N/Si ratio in the leaf tissue and shoot accumulation averaged 1.9 and 9.6, respectively, in both seasons, while the N/Si ratio in the root accumulation was 0.64. In the second season, Si appeared to be more soluble, or a residual effect was observed, and greater uptake was detected, which affected the N/Si ratio in leaves, shoots, and roots. The N/Si ratio decreased from 2.4 (2015/16) to 1.6 (2016/17) in leaves and 17.0 (2015/16) to 5.9 (2016/17) in shoot, but consequently, the N/Si ratio in roots increased from 0.53 (2015/16) to 0.75 (2016/17). The Si concentration in tissue (11.95 and 17.51 g Si $kg^{-1}$) was near the suitable concentration range for K (17–35 g $kg^{-1}$ D.M.) and above the suitable concentration for Ca (2.5–8.0 g $kg^{-1}$ of D.M.), which are the second and third nutrients most absorbed by maize according to Cantarella et al. [48]. The fact that a significant amount of Si is absorbed by maize suggests that this nutrient could have a more significant role in crop production than has been realized by researchers and deserves further investigation.

In this study, the use of Si had little to no quantifiable effect on maize development and grain yield. The Si benefits are more frequent in hyperaccumulator crops [34, 35], which contain $SiO_2$ concentrations above 5% of shoot dry matter [68, 69]. In addition, increased grain yield is unlikely when available Si in soil is above 10.0 mg $kg^{-1}$ [40, 70]. In this study, the available soil Si contents was close to this range (9.4 mg $kg^{-1}$ at 0–0.20 m and 10.2 at 0.20–0.40 m). In addition, more Si became available as straw decomposed during the growing season (13.1 kg $ha^{-1}$ Si, and 38.3 of C/N ratio, Table 2). Some studies have reported Si can be beneficial under biotic and abiotic conditions [71, 72]. For example, Galindo et al. [41] studying inoculation methods associated with Si application reported an average wheat grain yield increase of 6.7% when seed inoculation and Si application were performed. Guével et al. [71] observed that foliar applications of three different Si products (Kasil 26.5% $SiO_2$; Silamol 2% $SiO_2$ and 46% $K_2O$; and MRD-250 43.2% of $SiO_2$) did not promote wheat plant growth compared to the control treatment under greenhouse conditions, except when the plants were infected by

powdery mildew. The fact that adequate amounts of Si were present in the soil used for this study and the lack of a positive response to added Si suggest that little to no biotic or abiotic stress was present during the growing seasons studied.

## Conclusions

Inoculation with *A. brasilense* was found to increase N shoot and root accumulation, number of grains per ear, harvest index, NUE, and grain yield and also LCI when limestone was used. Yield increased by as much as 15% when plants were inoculated and fertilizer application was omitted and 10% when N was applied, showing the potential for nitrogen fixation by the PGPB used in this study. The utilization of Ca and Mg silicate as a Si source had small effects on plant development and grain yield and mostly increased Si uptake by the plant. The amount of Si removed was comparable to the amount of K removed and greater than the amount of Ca removed by maize grain. This study used an irrigated field which could have created conditions that hindered our ability to fully investigate the potential benefits of Si application. Therefore, studies conducted under conditions that challenge the crop in terms of stress, biotic or/and abiotic, are necessary to better understand the role of Si, applied alone or in combination with growth-promoting bacteria.

## Supporting information

**S1 Fig. Study area at the Selvíria, state of Mato Grosso do Sul, Brazil (20º22′S, 51º22′W, the altitude of 335 m above sea level).** Image obtained in Google Earth program. Google company (2019).
(TIF)

**S1 Raw Data.**
(XLSX)

## Author Contributions

**Conceptualization:** Fernando Shintate Galindo, Paulo Humberto Pagliari, Salatiér Buzetti, Marcelo Carvalho Minhoto Teixeira Filho.

**Data curation:** Fernando Shintate Galindo, Willian Lima Rodrigues, José Mateus Kondo Santini, Eduardo Henrique Marcandalli Boleta, Poliana Aparecida Leonel Rosa.

**Formal analysis:** Fernando Shintate Galindo, Paulo Humberto Pagliari, Willian Lima Rodrigues, José Mateus Kondo Santini, Eduardo Henrique Marcandalli Boleta, Poliana Aparecida Leonel Rosa.

**Funding acquisition:** Marcelo Carvalho Minhoto Teixeira Filho.

**Methodology:** Fernando Shintate Galindo, Paulo Humberto Pagliari, Salatiér Buzetti, Marcelo Carvalho Minhoto Teixeira Filho.

**Writing – original draft:** Fernando Shintate Galindo.

**Writing – review & editing:** Fernando Shintate Galindo, Paulo Humberto Pagliari, Salatiér Buzetti, Thiago Assis Rodrigues Nogueira, Edson Lazarini, Marcelo Carvalho Minhoto Teixeira Filho.

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
