## [Decision Letter · Decision Letter 0]

14 Oct 2019

PONE-D-19-24172

Can silicon applied to correct soil acidity in combination with Azospirillum brasilense inoculation improve nitrogen use in maize?

PLOS ONE

Dear Author,

Thank you for submitting your manuscript to PLOS ONE. After careful consideration, we feel that it has merit but does not fully meet PLOS ONE’s publication criteria as it currently stands. Therefore, we invite you to submit a revised version of the manuscript that addresses the points raised during the review process.

Thank you for submitting your manuscript to PLOS ONE. After careful consideration, we feel that it has merit, but is not suitable for publication as it currently stands. Therefore, my decision is "Major Revision." 

We invite you to submit a revised version of the manuscript that addresses all the comments raised by the reviewers.

We would appreciate receiving your revised manuscript by Nov 28 2019 11:59PM. To enhance the reproducibility of your results, we recommend that if applicable you deposit your laboratory protocols in protocols.io, where a protocol can be assigned its own identifier (DOI) such that it can be cited independently in the future. For instructions see: http://journals.plos.org/plosone/s/submission-guidelines#loc-laboratory-protocols

We look forward to receiving your revised manuscript.

Kind regards,

Dr. Umair Ashraf

Academic Editor

PLOS ONE

**Journal Requirements:**

http://www.scielo.br/scielo.php?lng=en&pid=S0006-87052018000300493&script=sci_arttext&tlng=en

https://mra.asm.org/content/6/20/e00393-18.long

https://www.mdpi.com/2223-7747/6/3/35/htm

In your revision ensure you cite all your sources (including your own works), and quote or rephrase any duplicated text outside the methods section. Further consideration is dependent on these concerns being addressed

3.  Our editorial staff has assessed your submission, and we have concerns about the grammar, usage, and overall readability of the manuscript.  We therefore request that you revise the text to fix the grammatical errors and improve the overall readability of the text before we send it for review. We suggest you have a fluent, preferably native, English-language speaker thoroughly copyedit your manuscript for language usage, spelling, and grammar.

If you do not know anyone who can do this, you may wish to consider employing a professional scientific editing service.  

Whilst you may use any professional scientific editing service of your choice, PLOS has partnered with both American Journal Experts (AJE) and Editage to provide discounted services to PLOS authors. Both organizations have experience helping authors meet PLOS guidelines and can provide language editing, translation, manuscript formatting, and figure formatting to ensure your manuscript meets our submission guidelines. To take advantage of our partnership with AJE, visit the AJE website (http://learn.aje.com/plos/) and enter referral code PLOS15 for a 15% discount off AJE services. To take advantage of our partnership with Editage, visit the Editage website (www.editage.com) and enter referral code PLOSEDIT for a 15% discount off Editage services. If the PLOS editorial team finds any language issues in text that either AJE or Editage has edited, the service provider will re-edit the text for free.

Please note that PLOS ONE does not copyedit accepted manuscripts and that one of our criteria for publication is that articles must be presented in an intelligible fashion and written in clear, correct, and unambiguous English (http://www.plosone.org/static/publication#language). If the language is not sufficiently improved, we may have no choice but to reject the manuscript without review.

**Additional Editor Comments (if provided):**

The manuscript "Can silicon applied to correct soil acidity in combination with Azospirillum brasilense inoculation improve nitrogen use in maize? is interesting but need revision as suggested by the reviewers.

i

**Comments to the Author**

1. Is the manuscript technically sound, and do the data support the conclusions?

Reviewer #1: Partly

Reviewer #2: Yes

Reviewer #3: Yes

2. Has the statistical analysis been performed appropriately and rigorously? 

Reviewer #1: Yes

Reviewer #2: Yes

Reviewer #3: Yes

3. Have the authors made all data underlying the findings in their manuscript fully available?

Reviewer #1: Yes

Reviewer #2: Yes

Reviewer #3: Yes

4. Is the manuscript presented in an intelligible fashion and written in standard English?

Reviewer #1: No

Reviewer #2: Yes

Reviewer #3: Yes

5. Review Comments to the Author

Reviewer #1: The manuscript entitled “Can silicon applied to correct soil acidity in combination with Azospirillum brasilense inoculation improve nitrogen use in maize” reported the interactive influence of silicon, Azospirillum brasilense inoculation and nitrogen application on maize growth, yield and NUE. Although, the theme of ms is interesting and falls in the scope of journal, yet I have serious concerns in various section of the draft.

The title of the article needs to be revised. It may cover your key findings.

Abstract is poorly structured.. Results are poorly drawn, conclude the section with your key findings.. I personally could not understand the treatment’s effects after reading the abstract section.

L36: optimized the N fertilization, increasing harvest index (HI), NUE and grain yield in 9.5, 19.3 and 5.5%, respectively????

Language needs substantial improvement. There are numerous grammatical and typo mistakes throughout the manuscript.

Introduction: Should be focused on the topic. Highlight research gap, and add clear cut objectives. I suggest adding recent studies on the interaction of Si and seed inoculation, Si and N, and N and seed inoculation.

Materials and Methods:

Experimental design: The experimental design was a completely randomized block design? To my knowledge, CRD is recommend only for controlled conditions.

L141: When N was applied?

Results: Too lengthy, can easily be brief. Parameters (e.g. yield and yield components) with similar trend can be combined. Be consistent regarding treatment description and the use of abbreviations. Better to add the numeric description of results (% variations) instead of just adding the data values for easy understanding of the readers. Various statements are confusing and unclear. It not enough to simply state the significant and non-significant effect of treatments..

Discussion should be merely based on the observed findings. Discussion on the interactive influence of silicon, Azospirillum brasilense inoculation and nitrogen application is poorly drawn. Conclusion: Report the key findings, consistent with the objectives.. It should not be the general summary. You may report some key genes.

Check whether the format of all references is according to the journal format.

Quality of Figures 2-6 needs to be improved.

Reviewer #2: The experiment has a certain novelty and workload. The manuscript is also well written and presented well. The data are presented clearly for the most part. Keeping in view of these study the follow justification needed:

1. More information about the cultivars should be provided in the Materials and Methods part including the breeding organization and why exactly these two cultivars had been chosen.

2. Accurate soil data in experiment site such as organic matter content, total nitrogen content, total potassium content and total phosphorus content should be provided.

3. The manuscript has a few grammatical errors which should be attended. A proofreading is suggested.

Reviewer #3: Please refer to the attached file for detailed point-wise comments and address/answer properly. Overall the manuscript is technically sound and conducted with viable research objectives and hypothesis. However, authors should need to consider related factors (i.e. N application in rotational crop) in expressing the study results.

6. PLOS authors have the option to publish the peer review history of their article (what does this mean?). If published, this will include your full peer review and any attached files.

Reviewer #1: No

Reviewer #2: Yes: Xiangru Tang

Reviewer #3: No

---

## [Author Response · Author response to Decision Letter 0]

30 Nov 2019

We would like to express our gratitude for the reviewers who took the time to provide such a thorough review of our manuscript. We believe that the changes suggested have made our manuscript much more direct and much easier to follow. We have addressed all of the concerns raised and provide a point by point answer on how we handled each comment provided. Our answers will be in italic and underlined right after each comment.

Because too many authors have revised this version it is possible that some language is still conflicting with some of our previous work. However, should you find this to be the case in any sentence of the revised document we will make every effort possible to fix it again. 

Again, our most sincere gratitude to you and the reviewers who took time from their busy schedule to help us making this manuscript a better paper. We hope that we have answered every inquiry to your satisfaction and also hope that you will find this version of publishable quality. Should you find that further work is needed we will also gladly do it in a timely manner. 

Very best,

Authors

PONE-D-19-24172

Can silicon applied to correct soil acidity in combination with Azospirillum brasilense inoculation improve nitrogen use in maize?

PLOS ONE

Dear Author,

Thank you for submitting your manuscript to PLOS ONE. After careful consideration, we feel that it has merit but does not fully meet PLOS ONE’s publication criteria as it currently stands. Therefore, we invite you to submit a revised version of the manuscript that addresses the points raised during the review process.

Thank you for submitting your manuscript to PLOS ONE. After careful consideration, we feel that it has merit, but is not suitable for publication as it currently stands. Therefore, my decision is "Major Revision."

We invite you to submit a revised version of the manuscript that addresses all the comments raised by the reviewers.

We would appreciate receiving your revised manuscript by Nov 28 2019 11:59PM. To enhance the reproducibility of your results, we recommend that if applicable you deposit your laboratory protocols in protocols.io, where a protocol can be assigned its own identifier (DOI) such that it can be cited independently in the future. For instructions see: http://journals.plos.org/plosone/s/submission-guidelines#loc-laboratory-protocols

• A rebuttal letter that responds to each point raised by the academic editor and reviewer(s). This letter should be uploaded as separate file and labeled 'Response to Reviewers'.

• A marked-up copy of your manuscript that highlights changes made to the original version. This file should be uploaded as separate file and labeled 'Revised Manuscript with Track Changes'.

• An unmarked version of your revised paper without tracked changes. This file should be uploaded as separate file and labeled 'Manuscript'.

We look forward to receiving your revised manuscript.

Kind regards,

Dr. Umair Ashraf

Academic Editor

PLOS ONE

Journal Requirements:

R: The authors appreciate the information. Thank you!

http://www.scielo.br/scielo.php?lng=en&pid=S0006-87052018000300493&script=sci_arttext&tlng=en

https://mra.asm.org/content/6/20/e00393-18.long

https://www.mdpi.com/2223-7747/6/3/35/htm

In your revision ensure you cite all your sources (including your own works), and quote or rephrase any duplicated text outside the methods section. Further consideration is dependent on these concerns being addressed

R: We have made our best attempts to address those issues. However, since most of this is our previous published work, we might have rephrased things back to similar language due to several authors revising the manuscript. Should you find any further occurrence of overlapping text we will gladly fix those right away.

We would also like to point out that after the revision, the turnitin program was used to verify similarity and only two articles showed similarity above 1%: 

1. Galindo FS, Rodrigues WL, Biagini ALC, Fernandes GC, Baratella EB, da Silva Junior CA, Buzetti S, Teixeira Filho MCM. Assessing forms of application of Azospirillum brasilense associated with silicon use on wheat. Agronomy. 2019;9:678.

2. Galindo FS, Teixeira Filho MCM, Buzetti S, Pagliari PH, Santini JMK, Alves CJ, et al. Maize yield response to nitrogen rates and sources associated with Azospirillum brasilense. Agronomy Journal. 2019;111:1985-1997.

Both studies were developed and published by our team and were properly cited and no data was reused, as can be verified by the lack of similarity mainly in the results and conclusions.

3. Our editorial staff has assessed your submission, and we have concerns about the grammar, usage, and overall readability of the manuscript. We therefore request that you revise the text to fix the grammatical errors and improve the overall readability of the text before we send it for review. We suggest you have a fluent, preferably native, English-language speaker thoroughly copyedit your manuscript for language usage, spelling, and grammar.

If you do not know anyone who can do this, you may wish to consider employing a professional scientific editing service. 

Whilst you may use any professional scientific editing service of your choice, PLOS has partnered with both American Journal Experts (AJE) and Editage to provide discounted services to PLOS authors. Both organizations have experience helping authors meet PLOS guidelines and can provide language editing, translation, manuscript formatting, and figure formatting to ensure your manuscript meets our submission guidelines. To take advantage of our partnership with AJE, visit the AJE website (http://learn.aje.com/plos/) and enter referral code PLOS15 for a 15% discount off AJE services. To take advantage of our partnership with Editage, visit the Editage website (www.editage.com) and enter referral code PLOSEDIT for a 15% discount off Editage services. If the PLOS editorial team finds any language issues in text that either AJE or Editage has edited, the service provider will re-edit the text for free.

R: Professor Dr. Paulo Pagliari from the University of Minnesota Soil, Water, and Climate department has provided an in-depth review of this version of the manuscript. In addition, a communication specialist (Mrs Emily Evans (eneperma@umn.edu)) who works closely with Dr. Pagliari has also provided a thorough revision of this version of the manuscript focusing on language.

Please note that PLOS ONE does not copyedit accepted manuscripts and that one of our criteria for publication is that articles must be presented in an intelligible fashion and written in clear, correct, and unambiguous English (http://www.plosone.org/static/publication#language). If the language is not sufficiently improved, we may have no choice but to reject the manuscript without review.

R: The authors appreciate the information. Thank you!

Additional Editor Comments (if provided):

The manuscript "Can silicon applied to correct soil acidity in combination with Azospirillum brasilense inoculation improve nitrogen use in maize? is interesting but need revision as suggested by the reviewers.

R: The authors appreciate the editor comments and restructured the entire paper as suggested by the reviewers. Thank you!

Comments to the Author

1. Is the manuscript technically sound, and do the data support the conclusions?

Reviewer #1: Partly

Reviewer #2: Yes

Reviewer #3: Yes

2. Has the statistical analysis been performed appropriately and rigorously?

Reviewer #1: Yes

Reviewer #2: Yes

Reviewer #3: Yes

 3. Have the authors made all data underlying the findings in their manuscript fully available?

Reviewer #1: Yes

Reviewer #2: Yes

Reviewer #3: Yes

 4. Is the manuscript presented in an intelligible fashion and written in standard English?

Reviewer #1: No

Reviewer #2: Yes

Reviewer #3: Yes

5. Review Comments to the Author

REVIEWER 1

Reviewer #1: The manuscript entitled “Can silicon applied to correct soil acidity in combination with Azospirillum brasilense inoculation improve nitrogen use in maize” reported the interactive influence of silicon, Azospirillum brasilense inoculation and nitrogen application on maize growth, yield and NUE. Although, the theme of ms is interesting and falls in the scope of journal, yet I have serious concerns in various section of the draft.

The title of the article needs to be revised. It may cover your key findings.

R: The authors thank all the great considerations of the reviewer. The title has been restructured and we followed your suggestion. 

Abstract is poorly structured. Results are poorly drawn, conclude the section with your key findings. I personally could not understand the treatment’s effects after reading the abstract section.

R: The abstract has been restructured and we followed your suggestion. Thank you.

L36: optimized the N fertilization, increasing harvest index (HI), NUE and grain yield in 9.5, 19.3 and 5.5%, respectively????

R: This sentence has been restructured and we followed your suggestion. 

Language needs substantial improvement. There are numerous grammatical and typo mistakes throughout the manuscript.

R: Two persons who are fluent in English (a US professor and a native speaker – communication specialist) have revised this version of the manuscript focusing on this aspect. We hope that this version is acceptable.

Introduction: Should be focused on the topic. Highlight research gap, and add clear cut objectives. I suggest adding recent studies on the interaction of Si and seed inoculation, Si and N, and N and seed inoculation.

R: We have done our best to address this comment. However, we feel that the introduction is already addressing what the reviewer would like to see in the introduction. Perhaps with the improved language our message will be easier to follow now.

Materials and Methods:

Experimental design: The experimental design was a completely randomized block design? To my knowledge, CRD is recommend only for controlled conditions.

R: This was a mistake by the student, the correct design was a RCBD, this has been better explained now. Thank you.

L141: When N was applied?

R: We have added the following information in the text: “The amount of fertilizer needed per plot was applied between the maize rows, on December 13, 2015, and December 10, 2016, when the plants were in the vegetative stage equivalent to V6 stage (with six leaves completely unfolded). After side dress application the experimental area was irrigated with 14 mm of water to minimize ammonia volatilization.”

Results: Too lengthy, can easily be brief. Parameters (e.g. yield and yield components) with similar trend can be combined. Be consistent regarding treatment description and the use of abbreviations. Better to add the numeric description of results (% variations) instead of just adding the data values for easy understanding of the readers. Various statements are confusing and unclear. It not enough to simply state the significant and non-significant effect of treatments.

Discussion should be merely based on the observed findings. Discussion on the interactive influence of silicon, Azospirillum brasilense inoculation and nitrogen application is poorly drawn. Conclusion: Report the key findings, consistent with the objectives. It should not be the general summary. You may report some key genes.

R: This was the section where most of the work was done to try and make the manuscript shorter and more direct to address all of the reviewers. We hope that this version has met the expectations of the reviewer.

Check whether the format of all references is according to the journal format.

R: Request made. Thank you.

Quality of Figures 2-6 needs to be improved.

R: To attend the reviewer and editorial team request, we have uploaded the figure files to the Preflight Analysis and Conversion Engine (PACE) digital diagnostic tool to ensure that figures meet PLOS One standard and requirements. Also, some figures have been replaced by tables to improve readability. Thank you.

REVIEWER 2

Reviewer #2: The experiment has a certain novelty and workload. The manuscript is also well written and presented well. The data are presented clearly for the most part. Keeping in view of these study the follow justification needed:

1. More information about the cultivars should be provided in the Materials and Methods part including the breeding organization and why exactly these two cultivars had been chosen.

R: The authors thank all the great considerations of the reviewer. We are a bit confused, but we think the reviewer refers to the bacteria strains used in the research. We have added more information regarding the strains used to answer this question.

2. Accurate soil data in experiment site such as organic matter content, total nitrogen content, total potassium content and total phosphorus content should be provided.

R: This information is presented in Table 1. Thank you.

3. The manuscript has a few grammatical errors which should be attended. A proofreading is suggested.

R: Two persons who are fluent in English (a US professor and a native speaker – communication specialist) have revised this version of the manuscript focusing on this aspect. We hope that this version is acceptable.

REVIEWER 3

Reviewer #3: Please refer to the attached file for detailed point-wise comments and address/answer properly. Overall the manuscript is technically sound and conducted with viable research objectives and hypothesis. However, authors should need to consider related factors (i.e. N application in rotational crop) in expressing the study results.

PONE-D-19-24172

Can silicon applied to correct soil acidity in combination with Azospirillum brasilense inoculation improve nitrogen use in maize? 

The titled manuscript was conducted to evaluate the interactive effects of Si and inoculation on maize yield and N uptake, the study provides good insights and the results will be useful in future on similar topic. However, there are few concerns which should be addressed (especially for data analysis) before the final consideration of manuscript, and the below comments should be responded properly and addressed carefully in revised draft. 

The title can be modified by replacing the term ‘nitrogen use’ with ‘nitrogen use efficiency’

R: The authors thank all the great considerations of the reviewer. The title has been restructured and we followed your suggestion. 

In abstract section, mostly the authors discussed about methods, and the results/conclusion part is too brief. Please describe the results more.

R: The abstract has been restructured and we followed your suggestion. Thank you!

If the authors have information for lime use practice by farmers in maize crop in Brazil’s maize region, please quote

R: The best management practice for lime application in these regions of Brazil is now cited in the revised version. Thank you for the recommendation.

L150 – given the ratio of compound fertilizer (08-28-16, N-P2O5-K2O), the blanket dose of N must be 30 kg ha–1, could you please justify?

R: The reviewer is right. We have corrected the blanket dose of N to 30 kg ha-1 and a justification is now provided in the text. 

In methods, please provide the general practice of fertilizer input rate in maize crop. And with reference to L126-128, authors are requested to add the details of fertilizer N input to the study site for at least previous crop before the maize plantation in 2015/16.

R: Information added. Thank you!

L170 – please mentioned the plant density in ‘m–2’ basis. In given dimensions, it should be 6.6 plants m–2. 

R: Information added. However, the plant density is 7.3 plants m–2 (space between rows is 0.45 m).

L188 – please provide the sampling frequency and time

R: The sampling took place only at once. Date of sampling is now reported. Thank you for the recommendation. 

L214 – please mention the statistical package name, which was employed to analyze the data? Also mention if the year effect was considered as random or fixed? Why the data was not analyzed across the years?

R: We have added the name of the statistical package used. The main reason we analyzed the data by year was because the crops grown before each of the maize crop studied differed in each season. In the first season maize was cropped in the winter of 2015 (June to September 2015) and in the second season wheat was grown in the winter of 2016. Therefore, we felt it would be more appropriate to analyze the data by year as opposed to combining both years.

L227 – please use proper sign for interaction (×) using ‘insert > symbol’ option in word program. Same is suggested for rest of the draft, and in Table 3.

Please rectify the errors on page 15 (L315) and page 19 (L432).

R: Change made. Thank you.

Discussion: please discuss the possibilities of factors affecting the study results due to other rotational wheat crop in the year 2016 (L127-128), what was the N rate in wheat crop? Why not the fallow period was observed between two maize crops?? Please consider this factor in discussing overall results.

R: Wheat was fertilized according to recommended rates throughout the entire area, so that is would not affect our results. In Brazil it is a common practice to grow a crop in the winter, short season crops usually, between the main summer crops. Therefore, to keep the results applicable to real farming conditions we opted for following practices that the farmers in the region adopt.

How the y-axis scale could be same for the temperature and precipitation in Fig. 1, please recheck by adding separate scale for precipitation.

R: The figure has been restructured and we followed your suggestion. Thank you!

From the figure data presented, it can be assessed that most of the times the differences between the two levels of a factor (N dose, lime sources, inoculation) are negligible or null. Please discuss this aspect in the results section or in the discussion. Secondly, mention the main effects of N dose affecting the grain yield or other salient parameters. The variance between each level shows no/slight differences. Please explain!!

R: We have tried to address this point throughout the text as we revised it. Please advise if more changes are needed

6. PLOS authors have the option to publish the peer review history of their article (what does this mean?). If published, this will include your full peer review and any attached files.

Do you want your identity to be public for this peer review? For information about this choice, including consent withdrawal, please see our Privacy Policy.

Reviewer #1: No

Reviewer #2: Yes: Xiangru Tang

Reviewer #3: No

 R: The authors appreciate the information and attend this request. Thank you!

---

## [Decision Letter · Decision Letter 1]

16 Jan 2020

PONE-D-19-24172R1

Can silicon applied to correct soil acidity in combination with Azospirillum brasilense inoculation improve nitrogen use efficiency in maize?

PLOS ONE

Dear Dr. Filho,

Thank you for submitting your manuscript to PLOS ONE. After careful consideration, we feel that it has merit but does not fully meet PLOS ONE’s publication criteria as it currently stands. Therefore, we invite you to submit a revised version of the manuscript that addresses the points raised during the review process.

ACADEMIC EDITOR: Reviewers raised concerns about the quality of the figures. Authors should improve the quality of the figures and upload again.

We would appreciate receiving your revised manuscript by Mar 01 2020 11:59PM. To enhance the reproducibility of your results, we recommend that if applicable you deposit your laboratory protocols in protocols.io, where a protocol can be assigned its own identifier (DOI) such that it can be cited independently in the future. For instructions see: http://journals.plos.org/plosone/s/submission-guidelines#loc-laboratory-protocols

We look forward to receiving your revised manuscript.

Kind regards,

Umair Ashraf

Academic Editor

PLOS ONE

Additional Editor Comments (if provided):

Authors have done with the comments/questions asked by the reviewers., however, reviewers and myself too also have concerns about the quality of figures. Authors should improve the resolution of the figures to make it clear.

Reviewers' comments:

Reviewer's Responses to Questions

**Comments to the Author**

1. If the authors have adequately addressed your comments raised in a previous round of review and you feel that this manuscript is now acceptable for publication, you may indicate that here to bypass the “Comments to the Author” section, enter your conflict of interest statement in the “Confidential to Editor” section, and submit your "Accept" recommendation.

Reviewer #1: All comments have been addressed

Reviewer #2: (No Response)

Reviewer #3: All comments have been addressed

2. Is the manuscript technically sound, and do the data support the conclusions?

Reviewer #1: Yes

Reviewer #2: Yes

Reviewer #3: Yes

3. Has the statistical analysis been performed appropriately and rigorously? 

Reviewer #1: Yes

Reviewer #2: Yes

Reviewer #3: Yes

4. Have the authors made all data underlying the findings in their manuscript fully available?

Reviewer #1: Yes

Reviewer #2: Yes

Reviewer #3: Yes

5. Is the manuscript presented in an intelligible fashion and written in standard English?

Reviewer #1: Yes

Reviewer #2: Yes

Reviewer #3: Yes

6. Review Comments to the Author

Reviewer #1: Although, the authors have responded well to all the comments raised by me, the quality of figures is too poor. Authors should check the Journal's guidelines regarding the minimum quality of figures

Reviewer #2: The expression of the manuscript has been substantially improved after modification.The author also supplemented relevant information in the paper for my questions.

Reviewer #3: Authors have addressed all the comments in revised version, and now the revised version looks good. However, authors didn't address the comment No. 9 during 1st review round. Please use proper symbol for interaction. Also the figures quality is even poor than the 1st submission, the minimum resolution of all figures should be 300dpi, please consider revising.

Thanks

7. PLOS authors have the option to publish the peer review history of their article (what does this mean?). If published, this will include your full peer review and any attached files.

Reviewer #1: Yes: Saddam Hussain

Reviewer #2: No

Reviewer #3: No

---

## [Author Response · Author response to Decision Letter 1]

29 Feb 2020

We would like to express our gratitude for the reviewers who took the time to provide such a thorough review of our manuscript. We believe that the changes suggested have made our manuscript much more direct and much easier to follow. We have addressed all of the concerns raised and provide a point by point answer on how we handled each comment provided. Our answers will be in italic and underlined right after each comment.

Again, our most sincere gratitude to you and the reviewers who took time from their busy schedule to help us making this manuscript a better paper. We hope that we have answered every inquiry to your satisfaction and also hope that you will find this version of publishable quality. Should you find that further work is needed we will also gladly do it in a timely manner. 

Very best,

Authors

PONE-D-19-24172R1

Can silicon applied to correct soil acidity in combination with Azospirillum brasilense inoculation improve nitrogen use efficiency in maize?

PLOS ONE

Dear Dr. Filho,

Thank you for submitting your manuscript to PLOS ONE. After careful consideration, we feel that it has merit but does not fully meet PLOS ONE’s publication criteria as it currently stands. Therefore, we invite you to submit a revised version of the manuscript that addresses the points raised during the review process.

ACADEMIC EDITOR: Reviewers raised concerns about the quality of the figures. Authors should improve the quality of the figures and upload again.

R: The authors appreciate the editor comments and restructured all the figures. Thank you!

We would appreciate receiving your revised manuscript by Mar 01 2020 11:59PM. To enhance the reproducibility of your results, we recommend that if applicable you deposit your laboratory protocols in protocols.io, where a protocol can be assigned its own identifier (DOI) such that it can be cited independently in the future. For instructions see: http://journals.plos.org/plosone/s/submission-guidelines#loc-laboratory-protocols

• A rebuttal letter that responds to each point raised by the academic editor and reviewer(s). This letter should be uploaded as separate file and labeled 'Response to Reviewers'.

• A marked-up copy of your manuscript that highlights changes made to the original version. This file should be uploaded as separate file and labeled 'Revised Manuscript with Track Changes'.

• An unmarked version of your revised paper without tracked changes. This file should be uploaded as separate file and labeled 'Manuscript'.

We look forward to receiving your revised manuscript.

Kind regards,

Umair Ashraf

Academic Editor

PLOS ONE

R: The authors appreciate the information. Thank you!

Additional Editor Comments (if provided):

Authors have done with the comments/questions asked by the reviewers., however, reviewers and myself too also have concerns about the quality of figures. Authors should improve the resolution of the figures to make it clear.

R: The authors appreciate the editor and reviewers comments and restructured all the figures. Thank you!

Reviewers' comments:

Reviewer's Responses to Questions

Comments to the Author

1. If the authors have adequately addressed your comments raised in a previous round of review and you feel that this manuscript is now acceptable for publication, you may indicate that here to bypass the “Comments to the Author” section, enter your conflict of interest statement in the “Confidential to Editor” section, and submit your "Accept" recommendation.

Reviewer #1: All comments have been addressed

Reviewer #2: (No Response)

Reviewer #3: All comments have been addressed

2. Is the manuscript technically sound, and do the data support the conclusions?

Reviewer #1: Yes

Reviewer #2: Yes

Reviewer #3: Yes 

3. Has the statistical analysis been performed appropriately and rigorously?

Reviewer #1: Yes

Reviewer #2: Yes

Reviewer #3: Yes 

4. Have the authors made all data underlying the findings in their manuscript fully available?

Reviewer #1: Yes

Reviewer #2: Yes

Reviewer #3: Yes

5. Is the manuscript presented in an intelligible fashion and written in standard English?

Reviewer #1: Yes

Reviewer #2: Yes

Reviewer #3: Yes

6. Review Comments to the Author

Reviewer #1: Although, the authors have responded well to all the comments raised by me, the quality of figures is too poor. Authors should check the Journal's guidelines regarding the minimum quality of figures

R: We have made our best attempts to address those issues. We replot all the figures as suggested by the reviewer. We hope that we have answered every inquiry to your satisfaction and also hope that you will find this version of publishable quality. Should you find that further work is needed we will also gladly do it in a timely manner. 

Thank you!

Reviewer #2: The expression of the manuscript has been substantially improved after modification. The author also supplemented relevant information in the paper for my questions.

R: The authors appreciate the reviewer comments. Thank you!

Reviewer #3: Authors have addressed all the comments in revised version, and now the revised version looks good. However, authors didn't address the comment No. 9 during 1st review round. Please use proper symbol for interaction. Also the figures quality is even poor than the 1st submission, the minimum resolution of all figures should be 300dpi, please consider revising.

Thanks

R: We have made our best attempts to address those issues. We replot all the figures as suggested by the reviewer. Also, we have changed all “x” by “×” in the interactions as suggested by the reviewer. We hope that we have answered every inquiry to your satisfaction and also hope that you will find this version of publishable quality. Should you find that further work is needed we will also gladly do it in a timely manner. 

Thank you!

7. PLOS authors have the option to publish the peer review history of their article (what does this mean?). If published, this will include your full peer review and any attached files.

Do you want your identity to be public for this peer review? For information about this choice, including consent withdrawal, please see our Privacy Policy.

Reviewer #1: Yes: Saddam Hussain

Reviewer #2: No

Reviewer #3: No

 R: The authors appreciate the information and attend this request. Thank you!

---

## [Editor Report · Decision Letter 2]

13 Mar 2020

Can silicon applied to correct soil acidity in combination with Azospirillum brasilense inoculation improve nitrogen use efficiency in maize?

PONE-D-19-24172R2

Dear Dr. Filho,

We are pleased to inform you that your manuscript has been judged scientifically suitable for publication and will be formally accepted for publication once it complies with all outstanding technical requirements.

With kind regards,

Dr Umair Ashraf

Academic Editor

PLOS ONE
---

## [Editor Report · Acceptance letter]

23 Mar 2020

PONE-D-19-24172R2 

Can silicon applied to correct soil acidity in combination with *Azospirillum brasilense* inoculation improve nitrogen use efficiency in maize? 

Dear Dr. Filho:

I am pleased to inform you that your manuscript has been deemed suitable for publication in PLOS ONE. Congratulations! Your manuscript is now with our production department. 

With kind regards,

on behalf of

Dr. Umair Ashraf 

Academic Editor

PLOS ONE